# Vascular Effects, Potential Pathways and Mediators of Fetal Exposure to Alcohol and Cigarette Smoking during Pregnancy: A Narrative Review

**DOI:** 10.3390/ijerph20146398

**Published:** 2023-07-19

**Authors:** Tammy C. Hartel, André Oelofse, Juléy J. A. De Smidt

**Affiliations:** Department of Medical Biosciences, Faculty of Natural Sciences, University of the Western Cape, Private Bag X17, Bellville, Cape Town 7530, South Africa

**Keywords:** maternal tobacco smoking, alcohol consumption, prenatal exposure, arterial health, children’s health, toxic effects

## Abstract

(1) Background: Programming of atherosclerosis results in vascular structure and function alterations, which may be attributed to fetal exposure to maternal tobacco smoking, alcohol consumption and several lifestyle factors in the first few years of life. This review aims to study the effects of teratogen exposure in utero on vascular dysfunction in offspring and consider mediators and pathways originating from the fetal environment. (2) Methods: Eligible studies were identified in the PubMed, Scopus and Web of Science databases. After the full-text screening, 20 articles were included in the narrative synthesis. (3) Results: The literature presents evidence supporting the detrimental effects of fetal exposure to tobacco smoking on vascular alterations in both human and animal studies. Alcohol exposure impaired endothelial dilation in animal studies, but human studies on both tobacco and alcohol exposure are still sparse. Reduction in nitric oxide (NO) bioavailability and alterations in the epigenome in infants through the upregulation of pro-oxidative and proinflammatory genes may be the common denominators. (4) Conclusion: While maternal smoking and alcohol consumption have more negative outcomes on the infant in the short term, several factors during the first few years of life may mediate the development of vascular dysfunction. Therefore, more prospective studies are needed to ascertain the long-term effects of teratogen exposure, specifically in South Africa.

## 1. Introduction

Throughout pregnancy, the placenta, umbilical cord and umbilical cord vessels play an important role in sustaining life and proper fetal growth. However, alterations in the fetal environment are directly or indirectly associated with in utero conditions, as the umbilical cord and its vessels are directly exposed to teratogens passing through the placenta [1]. Fetal exposures to these stressors, as stated by the Developmental Origins of Health and Disease (DOHaD) theory, can result in permanent changes in the structure, physiology and metabolism to adapt for fetal survival, but may lead to cardiovascular disease (CVD) risk factors later in life [2]. Balistreri CR (2020) provided an overview of recent evidence in the field of DOHaD of fetal programming of atherosclerosis. The programming of atherosclerosis results in vascular structure and function alterations, which can be attributed to fetal exposures, specifically during the embryonic and fetal stages and early postnatal life [3]. For example, fetal exposure to nicotine causes adverse outcomes, from low birth weight, oxidative stress and atherosclerotic lesions in the coronary vessels of infants to increased perinatal mortality [4]. Fetal alcohol exposure has also demonstrated consistent changes in blood vessels, such as endothelial dysfunction and stiffening, similar to that of nicotine [5,6,7]. Although research on the effects of alcohol on endothelial function in children is lacking, in a prospective twin study, alcohol consumption in the second trimester was strongly correlated with carotid–femoral pulse-wave velocity in nine-year-old children [8]. Furthermore, a recent prospective adult study reported direct negative effects of prenatal alcohol exposure on the reactive hyperemia index (RHI), with lower RHI scores indicating greater endothelial dysfunction [9]. 

Balistreri CR (2020) described the endothelium as a crucial target of fetal programming. This is due to intimal thickening, an adaptive physiological response to blood flow and vascular wall tension, which is more pronounced at sites near arterial bifurcations and branches and is prone to atherosclerotic lesion development [10]. These include the internal carotid artery, dorsal wall of the abdominal aorta and coronary arteries [10]. These arteries (for example, the carotid artery, with a vasculature that is straight and non-branching) can develop a more diffuse adaptive intimal thickening, a thickened intima of the arteries before the development of atherosclerosis [10,11].

Tobacco smoking and alcohol use during pregnancy are common, particularly in low socioeconomic regions in low-middle-income countries. Research studies report that approximately 40% of women report cigarette smoking in addition to heavy prenatal alcohol consumption during pregnancy [12]. However, the existing body of evidence on the potential pathological pathways of exposure to both alcohol and cigarette smoking exposure in utero on offspring vascular structure and function has yet to be synthesized. Therefore, this review aims to discuss the most recent evidence on the effects of teratogen exposures in utero on vascular dysfunction in offspring and consider the potential mediators in the development of atherosclerosis and pathways that may originate from the fetal environment. 

## 2. Materials and Methods

### 2.1. Literature Search

Eligible studies were identified in the PubMed, Scopus and Web of Science databases using specific search terms and Boolean combinations (Appendix A). The search found 1180 articles from PubMed (616), Scopus (1301) and Web of Science (260). The articles were then imported in RIS format to Endnote, and duplicates were removed. Thereafter, 32 full texts were screened in Endnote to select the most relevant articles to be included in the review. One reviewer screened and selected the most relevant articles. A total of 12 articles were included in the narrative synthesis, and 8 were identified through reference searching (Figure 1).

### 2.2. Eligibility Criteria

Observational studies, including human research studies and animal studies investigating the effects of maternal cigarette smoking and/or alcohol consumption during pregnancy on the fetus and vascular dysfunction in children or offspring, were considered eligible. Studies focusing on vascular dysfunction that included structural alterations, such as carotid intima–media thickness (cIMT) and atherosclerotic plaques, and functional alterations, such as flow-mediated dilation (FMD), arterial stiffness and pulse-wave velocity (PWV), were considered eligible. Animal studies included all types of animals with male and female offspring. Human studies included children from birth to 13 years old. Twin studies were excluded. Reviews, clinical trials, books, documents, and surveys were excluded from the synthesis, but all epidemiological studies on humans and animals were included to support the body of evidence. The language was restricted to English. However, there was no limitation on the year of publication and location.

### 2.3. Graphical Bibliographical Analysis of Literature Search

Figure 2 presents a graphical representation of all search results from PubMed, SCOPUS and Web of Science. The search results were saved and exported from Endnote 20.5 to Zotero™ (Version 6.0.13), where references were checked for duplicates. The citations were exported from Endnote as a reference manager file and imported to VOSviewer™ (Version 1.6.18), where the map was created. The diagram was normalized using LinLog/modularity. The diagram shows the most used keywords in articles in the searched databases and their link to one another. It is, therefore, a graphical representation of recent trends and growth in knowledge in the field of DOHaD. The map includes keywords that had a minimum of five occurrences in an article and the 106 keywords with the greatest strength of co-occurrence with other terms. The size of the circles indicates the commonality of each term and the line thickness and popularity of the key terms. The map displays two clusters representing two main themes (red and green) with closely related concepts.

## 3. Results

The current review identified twenty studies, of which ten studies reported evidence from humans and ten from animal studies as presented in Table 1 and Table 2, respectively. Based on the identified studies, evidence exists supporting the detrimental effects of fetal exposure to alcohol and tobacco smoking on vascular alterations. Section 3 presents the body of evidence from human and animal studies. It explores the relationship between fetal exposure to maternal smoking and alcohol consumption and vascular dysfunction in offspring.

### 3.1. Teratogen Exposure during Pregnancy and Vascular Dysfunction

Research studies have suggested that nicotine exposure influences vascular structure, leading to atherosclerotic plaque formation [2,4,13]. For example, one study reported a significantly greater aortic intima–media thickness (aIMT) in neonates born to smoking mothers during pregnancy [14]. In contrast, other studies reported no significant differences in cIMT and aIMT when comparing infants with exposure to maternal smoking to infants in the control group, with no associations reported [15,16]. However, as children grew older, studies observed a significantly thicker cIMT (18.8 µm, *p* = 0.04) in five-year-old children exposed to maternal smoking in utero [17], but also in children exposed to both maternal smoking and alcohol use in utero (*p* = 0.008) [7]. There remains a lack of sufficient evidence to make a concluding statement on the vascular effects of alcohol, as the most studied effects of alcohol consumption in utero are on brain development, memory, behavior abnormalities and learning impairment. 

Apart from human studies, animal studies administering nicotine similar to plasma nicotine levels of moderate human smokers reported potential negative effects [18,19]. A study on rats exposed to nicotine (1 mg/kg/day) during pregnancy and lactation observed an abnormal alignment of the abdominal aorta wall with an irregular arrangement of the smooth muscle cells within the tunica media [20]. This irregular alignment predisposed the rats to atherosclerotic plaque formation [20]. In addition, nicotine exposure also heightened oxidative stress in rats, supporting the concept of fetal programming of vascular dysfunction [21]. Therefore, with nicotine exposure, the aorta seemed to be the most common site in rat studies, with a trend of increased aIMT and decreased endothelium-dependent vasodilation in offspring exposed to nicotine treatment during pregnancy [18,22].

On the other hand, when studying the effect of fetal alcohol exposure, Ramadoss and Magness (2012) [23] discussed the difficulties of studying the underlying mechanisms of alcohol, despite alcohol being a simple molecule, due to the diverse effects of alcohol, depending on the duration of alcohol use during pregnancy (a trimester vs. throughout pregnancy) and acute, chronic or binge drinking, or the dosage. For example, animal studies showed that binge drinking resulted in higher blood alcohol concentrations and greater deficits compared to continuously administered high dosages of alcohol. In a baboon model, Tobiasz et al. (2018) [24] studied fetal vascular function with blood alcohol concentrations in pregnant baboons which qualified as binge drinking. The authors were able to standardize the timing, dosage and patterns of alcohol exposure, and results showed that fetal alcohol exposure in the second trimester caused fetal vasodilation of the cerebral arteries, measured by flow velocity waveforms [24]. In addition, rat models demonstrated that moderate as well as binge alcohol consumption resulted in vascular dysfunction in the uterine artery by impeding the vasodilation mediated by the endothelial nitric oxide synthase (eNOS) pathway, which may impact fetal development [25,26]. 

Therefore, the insults of alcohol exposure in utero on vascular function as well as fetal growth may have different time courses, of which the underlying mechanism is uncovered. Trends suggest that fetal exposure to nicotine results in vascular alterations, specifically in the carotid artery in humans, whereas both nicotine and alcohol exposure in rats has detrimental effects on the aortic artery and uterine arteries [18,20,25] but also affects other cardiac characteristics such as increasing inflammatory factors (tumor necrosis factor (TNF)-α, endothelin-1 and nuclear factor (NF)-κ, and intercellular adhesion molecule (ICAM)-1) and disruption in the nitric oxide (NO) synthesis pathway [19,26]. Interestingly, exposure to alcohol in pregnant rats had detrimental effects on vascular function, with several studies reporting a decrease or impairment in endothelium-dependent vasodilation [21,22,25,26,27,28]. Therefore, strong evidence exists that supports the concept of fetal programming of vascular dysfunction in offspring exposed to alcohol exposure in utero and has been discussed previously [6,29,30].

### 3.2. Teratogen Exposure during Pregnancy and Altered Lipid Metabolism

A recent study examining the association between maternal tobacco smoking and cardiometabolic health in children found significantly higher triglyceride concentrations in 10-year-old children exposed to continued maternal tobacco smoking compared to non-exposed children [31]. However, total cholesterol was not associated with tobacco smoking in utero [31]. One study on neonates reported significantly lower high-density lipoprotein (HDL) concentrations in infants exposed to maternal smoking during pregnancy but no significant differences in LDL-cholesterol, triglycerides and total cholesterol [16]. Cajachagua-Torres and colleagues observed sex differences; the associations of fetal tobacco exposure with lipid concentrations and blood pressure were stronger among boys, whereas the association with glucose outcomes seemed stronger in girls [31].

In addition, altered lipid metabolism due to ethanol exposure in utero has been reported in a systematic review, specifically hypercholesterolemia and/or dyslipidemia in offspring with in utero alcohol exposure [32,33]. Although the results showed altered lipid metabolism despite the dose or timing of alcohol exposure, Akison et al. (2019) suggested that a dose-dependent relationship exists between in utero alcohol consumption and the development of dyslipidemia in offspring [32]. Although in adult studies, patients with prenatal alcohol exposure had increased metabolic abnormalities, such as low HDL-cholesterol (31.9%), elevated triglycerides (34.5%) and overweight/obesity (64.9%) [33]. However, males had a higher incidence of metabolic abnormalities (61.2%) despite their low BMI compared to females (35.9%) [33]. 

In animal studies, Weeks et al. (2020) used a zebrafish model for prenatal alcohol exposure, reporting prenatal alcohol exposure as a risk factor for diet-induced obesity in male zebrafish. This shows evidence that maternal alcohol consumption may play an important role in dyslipidemia and, therefore, cIMT in the offspring. Thus, many animal studies have presented a body of evidence to support the DOHaD and programming of the development of atherosclerosis which recommended longitudinal studies be conducted to study the long-term effects of in utero exposure to alcohol.

**Table 1 ijerph-20-06398-t001:** Summary of the vascular effects in human studies.

Reference	Year, Country	Sample	Study Design	Analysis Methods	Outcome
Anderson, Walker and Stender. [34]	2004, Copenhagen	74 pregnant women and infants (Smoking = 30, Control = 44)	Prospective pregnancy/birth cohort study	Endothelial cells of fetuses’ umbilical veins were isolated after birth, and eNOS activity was studied by an eNOS immunoassay.	The eNOS activity in the umbilical veins of fetuses exposed to maternal smoking was significantly reduced (*p* = 0.006) by 40%, with a 32% lower concentration of eNOS, suggesting endothelial dysfunction and reduced fetal growth.
Gunes et al. [14]	2007, Kayseri, Türkiye	28 term neonates with smoking mothers and 28 term neonates with non-smoking mothers	Prospective pregnancy/birth cohort study	Aortic artery IMT was measured in newborns using high-resolution ultrasound.	The mean aIMT and weight-adjusted aIMT were significantly greater in neonates born to smoking mothers during pregnancy compared to neonates in the control group. Maternal smoking did not significantly decrease insulin-like growth factor I (IGF-I) and IGFBP-3 levels in their infants.
Geelhoed et al. [35]	2011, Rotterdam, The Netherlands	1120 mothers and their two-year-old children (≥5 cigarettes per day = 72), (<5 cigarettes per day = 68)	A prospective birth cohort study	Pulsed Doppler ultrasound assessed fetal and placental arterial resistance characteristics during the third trimester. At two years old, cardiac measurements were evaluated using M-mode and Doppler echocardiograms.	Continued smoking during pregnancy (≥5 cigarettes per day) was associated with increased fetal arterial resistance in the umbilical (*p* < 0.01), uterine (*p* < 0.01) and middle cerebral arteries (*p* < 0.05), and with a decrease in blood flow (*p* < 0.05) and diameter (*p* < 0.01) in the ascending aorta in the third trimester. Increased fetal umbilical artery resistance was also associated with a reduced aortic root diameter at two years old.
Iwashima et al. [15]	2012, Hamamatsu, Japan	96 mothers and newborns	Prospective pregnancy/birth cohort study	Abdominal aortic artery IMT in newborns using high-resolution ultrasound.	No associations were investigated between maternal smoking and newborn aIMT due to limitations. However, there were no significant differences in newborn aIMT when comparing newborns exposed to smoking mothers vs. non-smoking mothers.
Geerts et al. [17]	2012, Utrecht, Netherlands	259 five-year-old children and their parents (maternal smoking = 15, non-smoking = 244)	A prospective birth cohort study	cIMT and arterial wall distensibility were measured using ultrasound.	Children born to maternal smoking during pregnancy had significantly thicker cIMT (18.8 μm) at five years old (*p* = 0.04), with 15% significantly lower arterial wall distensibility. Maternal smoking was not associated with cIMT after adjusting for the child’s growth patterns.
Taal et al. [36]	2013, The Netherlands	5565 mothers and six-year-old children (non-smoking n = 4159, continued smoking n = 912, stopped smoking n = 494)	A prospective birth cohort study	Blood pressure, carotid–femoral pulse-wave velocity and left cardiac structures and function were assessed.	No association was observed between maternal cigarette smoking during pregnancy with systolic blood pressure (SBP), carotid–femoral PWV and left cardiac structures. Maternal smoking of 10 or more cigarettes per day was associated with a higher fractional shortening in childhood. A dose–response association was observed between the number of cigarettes smoked during the third trimester and diastolic blood pressure (DBP) in six-year-old children.
Dugmonits et al. [37]	2019, Hungary	113 mothers and neonates (non-smoking n = 62, smoking n = 51)	Ex vivo study	Red blood cells were isolated from fetal umbilical cord artery samples for morphological and molecular studies.	Circulating red blood cells in fetuses exposed to maternal smoking presented with impaired activation of nitric oxide synthase 3 (IGFBP-3), regardless of the phenotypic appearance, as well as an upregulation of the Arginase 1 (ARG1) enzyme, a molecule in the NOS3-NO pathway. The association between decreased NOS3 and maternal smoking was not investigated.
De Smidt et al. [7]	2021, South Africa	500 five-year-old children (control n = 146, smoking n = 167, alcohol n = 33, dual-exposed n = 154)	Prospective pregnancy/birth cohort	Ultrasound measurements of the aorta and carotid arteries IMT were performed at the age of five years	cIMT was significantly higher in children exposed to both alcohol and cigarette smoking during pregnancy compared to those not exposed (*p* = 0.008). Dual exposure was associated with higher right cIMT (*p* < 0.01).
Turan et al. [16]	2021, Turkey	74 mothers and term neonates aged 0–30 days (smoking n = 40, control n = 34)	Prospective pregnancy/birth cohort	Carotid IMT was measured by standard ultrasound examination, and fasting blood cholesterol was determined by an automatic chemistry analyzer.	There were no differences in carotid IMT and low-density lipoprotein (LDL)-cholesterol, triglycerides and total cholesterol between neonates exposed to maternal smoking (n = 40) and neonates in the control group (n = 34). However, neonates exposed to maternal smoking had significantly lower HDL levels (39.1 ± 17.8 mg/dL; *p* = 0.021).
Monasso et al. [38]	2022, The Netherlands	4639 mothers and 10-year-old children (control n = 3180, smoking n = 627, quit smoking n = 359)	Prospective pregnancy/birth cohort	Common carotid artery intima–media thickness and distensibility were assessed using standard ultrasound protocols.	Continued maternal tobacco smoking was not associated with arterial health markers such as cIMT and distensibility at the age of 10 years old.

Note: eNOS—endothelial nitric oxide synthase; IMT—intima–media thickness; aIMT—aortic intima–media thickness; IGF-I—insulin-like growth factor I; IGFBP-3—insulin-like growth factor-binding protein-3; cIMT—carotid intima–media thickness; SBP—systolic blood pressure; PWV—pulse-wave velocity; DBP—diastolic blood pressure; ARG1—arginase 1; NOS3—nitric oxide synthase-3; NO—nitric oxide; HDL—high-density lipoprotein; mg/dL—milligrams per deciliter.

**Table 2 ijerph-20-06398-t002:** Summary of the vascular effects in animal studies.

Reference	Year, Country	Sample Size	Analysis Methods	Outcome
Turcotte et al. [27]	2002, North Dakota	31 female pregnant Sprague Dawley rats and offspring (control = 18, ethanol exposed = 13)	Vascular function was studied in fetal alcohol syndrome rats by evaluating force-generating capacity of thoracic aorta segments isolated from adult rat offspring.	Ethanol treatment resulted in altered vascular contractile function in aortic segments, whether the responses were endothelium-dependent or endothelium-independent.
Xiao et al. [22]	2007, California, United States	17 pregnant Sprague Dawley Rats (control = 9, nicotine-treated = 8)	Isolated aortic from adult male and female offspring at three months old.	Prenatal nicotine exposure decreased aortic endothelium-dependent vasodilation induced by acetylcholine (Ach) in male rat offspring. However, it increased endothelium-dependent relaxation in female offspring. Nicotine treatment resulted in vascular alterations differently in each gender.
Xiao et al. [21]	2011, United States	25 pregnant Sprague Dawley rats and male rat offspring	Aortas were isolated, contraction studies were performed and Western blot analysis was used to determine protein expression of NADPH oxidase (Nox2 and Nox4). Oxidative damage was determined by measuring malondialdehyde levels, superoxide production and superoxide dismutase.	Aortic relaxations induced by Ach were impaired in rat adult offspring exposed to nicotine treatment in pregnant rats. Nicotine exposure resulted in increased oxidative stress and heightened Nox2-dependent mediated hypertensive reactivity. Therefore, prenatal nicotine exposure in a rat model supports the concept of fetal programming of vascular dysfunction through heightened oxidative stress.
Gunes et al. [18]	2011, Kayseri, Türkiye	25 female adult white Sprague Dawley and offspring (control n = 5, nicotine group A n = 10, nicotine group B (n = 10)	Using histopathology, the abdomen aortic IMT in rat pups was studied at 45 days of age.	The aIMT was significantly higher in rat pups in both nicotine groups exposed to the nicotine treatment compared to the controls (*p* < 0.001). No significant difference in aIMT when comparing the two nicotine groups that received different concentrations of nicotine (nicotine group A: 6 mg/kg/day and nicotine group B: 3 mg/kg/day).
Parkington et al. [28]	2014, Australia	Twelve female sheep (ewes) and their fetuses, six exposed and six non-exposed fetuses	Arteries were isolated on day 134 of pregnancy for vascular function tests using a wire myograph. Arterial stiffness was assessed using a pressure myograph.	Daily ethanol exposure during late pregnancy reduced coronary arteries’ endothelium-dependent vasodilatation sensitivity by 10-fold without fetal growth restriction. Ethanol exposure was also significantly associated with reduced endothelial NO synthase mRNA. Arterial stiffness was increased in all isolated arteries (coronary, renal, mesenteric, psoas muscle and cerebral).
Subramanian et al. [26]	2015, United States	12 Sprague Dawley rats and fetuses on gestational day 18	Vascular studies were performed using wire myography to investigate uterine artery function to vasodilators and vasoconstrictors in a pregnant rat model.	Moderate alcohol exposure resulted in impaired acetylcholine-mediated and endothelium-dependent vasodilation in uterine arteries of pregnant mothers in the absence of growth deficits.
Naik et al. [25]	2016, United States	16 Sprague Dawley rats and fetuses (control = 8, alcohol exposed = 8)	The uterine artery was isolated on gestational day 20, and functional tests were performed using dual-chamber arteriography.	Daily binge alcohol exposure resulted in impaired endothelial vasodilation in the primary uterine artery by impeding the eNOS pathway.
Tobiasz et al. [24]	2018, United States	18 pregnant baboons (control = 9, alcohol exposed = 9)	Pregnant baboons were treated with alcohol three times in the 2nd semester. Fetal cardiovascular parameters were assessed using Doppler ultrasound.	Fetal cerebral blood flow (peak systolic velocity) decreased during alcohol exposure, but no significant differences were found in fetal cardiovascular indices at term. Therefore, alcohol affected vascular function, but no persistent changes were observed.
Alfourti et al. [20]	2019, Libya	Pregnant Wister rats (sample size unknown)	Pregnant rats were treated with nicotine (1 mg/kg/day in 1 mL). The abdominal aorta was isolated for histological studies.	The three tunica layers of the abdominal aorta showed irregular alignment in rats exposed to nicotine compared to regular alignment in the control group.
Soraya et al. [19]	2022, Iran	20 Wister rats and their male rat offspring	Systemic hemodynamic measurements were analyzed in offspring on postnatal day 90, utilizing waveform contour analysis. Histology of the aortic wall and level of inflammatory factors of the aorta were assessed at postnatal days 21 and 90.	Aortic wall thickness significantly increased in the ethanol-exposed group at postnatal days 21 and 90. Inflammatory factors such as tumor necrosis factor (TNF)-α, endothelin-1 and nuclear factor (NF)-κ, and intercellular adhesion molecule (ICAM)-1 significantly increased in the ethanol-exposed group (*p* < 0.001), predisposing the aorta to atherosclerosis. SBP, DBP, mean arterial pressure (MAP) and dicrotic pressure were significantly higher in rat offspring with ethanol exposure.

Note: Ach—acetylcholine; NADPH—nicotinamide adenine dinucleotide phosphate; Nox—NADPH oxidase; IMT—intima-media thickness; aIMT-; mg/kg/day—milligrams per kilogram per day; mRNA—messenger ribonucleic acid; NO—Nitric oxide; eNOS—endothelial nitric oxide synthase; ml—milliliter; (TNF)-α—tumor necrosis factor-alpha; (ICAM)-1—intercellular adhesion molecule-1; SBP—systolic blood pressure; DBP—diastolic blood pressure; MAP—mean arterial pressure.

In summary, based on the literature search, it is possible that several socioeconomic and lifestyle factors during childhood, in addition to the detrimental effects of maternal alcohol and cigarette smoking on the infant, are likely to influence and progress the development of vascular dysfunction, therefore increasing the risk of atherosclerosis later in life, as shown in the diagram in Figure 3.

## 4. Discussion

This review aimed to synthesize the most recent evidence on the effects of fetal exposure to maternal tobacco smoking and alcohol use and vascular dysfunction in offspring, and to consider the potential pathways and mediators in the development of vascular dysfunction that may originate from the fetal environment. Herein, we will discuss potential pathways and mediators in the development of vascular dysfunction in fetuses and children based on the current literature.

Firstly, the teratogenic effects of alcohol are specifically relevant in pregnancy as the fetal alcohol concentrations are almost equivalent to the maternal alcohol concentration, exposing the fetus to the teratogenic effects of alcohol before the toxin is eliminated [39]. During gestation, the fetus is more prone to alterations in its oxidative metabolic pathway due to several reasons. Alcohol’s toxicity on the fetus is mainly due to the production of reactive oxygen species (ROS) through the conversion of ethanol to acetaldehyde by the enzyme CYP2E1 protein, which becomes the major metabolic pathway during ethanol exposure. In fetal alcohol syndrome (FAS), ROS leads to placenta vasoconstriction, brain lipid peroxidation, mitochondrial damage, apoptosis, and inhibition of cofactors that are essential for fetal development [40].

As stated previously, studies have proposed that alcohol causes increased oxidative stress, which leads to redox alterations [6,41]. Oxidative stress is an imbalance in reactive species such as ROS and reactive nitrogen species (RNS), leading to oxidative damage in macromolecules [6,39]. Oxidative stress also decreases NO synthesis, a vasodilator which inhibits platelet aggregation and adhesion and, therefore, atherosclerosis [42]. Toda and Ayajiki (2010) [5] reported that low concentrations of alcohol increase NO production in human endothelial cells, whereas high concentrations induce endothelial dysfunction and apoptosis. 

In contrast, Parkington, Coleman and Wintour (2010) concluded that alcohol enhances NO production and has a direct effect on vascular smooth muscle by increasing the density of Endothelin B (ET_B_) receptors on endothelial cells, which in turn promotes NO release. Although, when endothelial cells are persistently challenged with an inhospitable environment, it can lead to oxidative stress, endothelial dysfunction and a decrease in NO bioavailability [30].

However, there remains a lack of data from human clinical studies to conclude the effects of alcohol exposure on vascular function in children. In this review, only one study included both alcohol and nicotine exposure in utero and observed an increase in cIMT in 5-year-old children, suggesting that exposure to both teratogens may have compounding effects on the intima–media thickness compared to nicotine only [7]. In addition, a recent study by Marianian et al. (2019) [41] reported an association between small amounts of alcohol consumed during pregnancy and dysfunction of the lipid peroxidation–antioxidant defense (LPO-AOD) system in both women and their infants. This suggests that alcohol exposure may cause oxidative stress through an imbalance in the LPO-AOD system [41]. Furthermore, a cross-sectional study on South African children in rural and urban areas reported an association between oxidative stress and vascular dysfunction. Common markers to measure oxidative stress: thiobarbituric acid reactive substances (TBARS) and 8-hydroxyl-2-deoxy guanosine (8-OHdG), resulting in cell membrane and DNA damage [43]. Findings from this study by Matjuda et al. (2021) showed that the marker, 8-OHdG, was correlated with asymmetric dimethylarginine (ADMA) (an eNOS inhibitor) and increased with increasing ADMA quartiles [43,44]. 8-OHdG was also found to be a predictor of endothelial dysfunction in children (as measured by PWV) [43]. Thus, increasing ADMA decreases NO bioavailability as it inhibits eNOS by competing with L-arginine, resulting in impaired NO production [43,44].

Like alcohol exposure, nicotine also compromises NO production. For example, fetuses exposed to maternal smoking showed significantly reduced eNOS activity (by 40%) in the umbilical veins (*p* = 0.006), with a 32% lower concentration of eNOS compared to non-exposed fetuses [34]. A recent study on endothelial dysfunction in umbilical cord arteries and veins originating from neonates exposed to heavy maternal tobacco smoking detected an increase in the formation of ROS and upregulation of the NOS2-NO-producing pathway [1]. Zahorán et al. (2021) [1] suggest that the umbilical vein is more affected in terms of structural and molecular damage compared to the artery, as detected by ultrastructural changes. Due to the lack of innervation, dilation of the umbilical cord vessels relies primarily on the bioavailability of NO produced by the activated NOS2 pathway. Therefore, any structural damage to the vessel and plasma membrane will lead to molecular damage, such as compromised NO production. Consequently, alternative compensatory pathways, including the NOS2 and xanthine oxidoreductase (XOR), are upregulated in the vein to provide an alternative source for bioavailable NO. However, the up-regulation of this pathway (NOS2) will lead to cytokine stimulation and an increase in ROS [1]. This will further exhibit macromolecular and cellular damage and eventually lead to vascular dysfunction. Furthermore, an in vitro study by Chaturvedi et al. (2015) suggests that nicotine affects the structural and functional characteristics of blood vessels and endothelial cells by stimulating DNA synthesis and endothelial cell proliferation [13]. These changes may lead to vascular dysfunction and atherosclerotic plaque formation [13].

Apart from oxidative stress, epigenetics is also particularly relevant in studies investigating the origins of vascular dysfunction and cardiometabolic disease in offspring exposed to alcohol or nicotine in utero. For example, alcohol exposure in utero has consistently demonstrated vascular remodeling, endothelial dysfunction and stiffening, which may be mediated by the control of gene expression, which, through their aberrant regulation, may lead to CVD later in life [6]. Studies suggest that in utero exposure, albeit nicotine or alcohol, can cause significant changes in deoxyribonucleic acid (DNA) methylation in the fetus [2,30,45]. Ivorra et al. (2015) studied DNA methylation patterns in normal-weight infants born at term and exposed to maternal smoking in utero; the author observed statistically significant differences in 31 CpG sites in association with 25 genes. In addition, higher methylation levels were observed in 90.3% of loci located in the CpG islands in the exposed group [46]. Therefore, exposure to tobacco smoking in utero may alter the epigenome in infants, despite fetal growth restriction [46]. Equally important, tobacco smoking during pregnancy may also alter the placental genome, as 20 CpG sites associated with maternal smoking during pregnancy were identified in 194 placental samples, of which 5 CpG sites were associated with gestational age at birth and 6 with birthweight z scores [47]. Specifically, higher DNA methylation levels (at cg17823829) and lower DNA methylation levels (at cg26843110) in the placenta were correlated with shorter gestational age at birth [47]. Therefore, suggesting that nicotine or alcohol exposure affects placental gene expression involved in the programming of lipid and glucose metabolism, energy expenditure and uptake, inflammation and regulation of blood pressure, and fetal growth regulation, which are important biological factors in the pathogenesis of cardiometabolic syndrome [47,48]. Everson et al. (2021) also identified pathways critical to placental and fetal development, including the aryl hydrocarbon receptor (Ahr) pathway, recognized for responding to environmental stressors and regulating immune activity. T helper 17 (Th17) cell differentiation, one of the pro-inflammatory immune cell pathways identified in these maternal-smoking-associated genes, can also induce inflammation and oxidative stress within the placenta [47].

To illustrate, a recent study suggested that exposure to maternal smoking induced long-lasting epigenetic changes which persisted into adolescence [49]. The results by Rauschert et al. (2019) showed significant associations between CpG methylation and maternal tobacco smoking, as well as cardiometabolic risk factors in adolescents, specifically between the cg00253568 (FTO gene) and cg00213123 (CYP1A1 gene) CpGs in association with triglycerides, HDL cholesterol and diastolic blood pressure in adolescents, after a Bonferroni correction [49]. Constantino and colleagues (2018) also demonstrated a link between epigenetic remodeling and oxidative stress, insulin resistance and endothelial dysfunction through the upregulation of pro-oxidant and pro-inflammatory genes [50]. The beforementioned studies strongly suggest the persistent role that epigenetics may play in the pathogenesis of cardiometabolic features.

From a histopathological perspective, Soraya and colleagues (2022) propose that inflammatory cytokines mediated induced aorta abnormalities in male rat offspring exposed to ethanol in utero. The transcriptional factor of the nuclear factor kappa B signaling pathway (NF-kB), in particular, may be attributable to several gene expressions involved in inflammatory responses, cell growth and adhesion molecules, including ICAM-1, Interleukin-6 (IL-6), Tumor necrosis factor-alpha (TNF-α) and beta-selectin [19]. This theory is supported by evidence of significantly higher levels of NF-kB, ICAM-1 adhesion molecules and TNF-α in the aortic tissue of ethanol-exposed rat offspring [19]. Therefore, ROS generation in the endothelial cell through activation of NADPH oxidase, following activation of TNF, seems to regulate the expression of adhesion molecules. In rats exposed to nicotine in utero, increased ROS in vascular tissue significantly contributes to fetal programming of vascular dysfunction [21]. 

Nicotine exposure increases vascular superoxide production by upregulation of Ang II/AT1 R-mediated Nox2 signaling in rat offspring. This plays an important role in the programming of vascular reactivity, supporting the theory of heightened oxidative stress [21].

### 4.1. Potential Mediators in Vascular Dysfunction

It is noteworthy to acknowledge that maternal smoking and alcohol use during pregnancy has mediating effects on vascular function in children. Potential mediators include birth size, catch-up growth and obesity, and elevated blood pressure. Maternal smoking may have an indirect effect on cIMT through birth size. However, analyses of the mediation effect of birth size are needed to confirm this hypothesis [51]. A recent systematic review showed a consistent association between small-for-gestational-age (SGA) and higher cIMT in children, with a stronger association when restricted to studies with high-quality cIMT measurements [51]. 

In addition, children born SGA have been associated with rapid weight gain and childhood adiposity [12]. A study examining the relationship between body mass index (BMI) during critical periods of life and arterial health in children reported an association between higher peak weight velocity indicating maximum infant growth rate, BMI at maximum adiposity and increased cIMT at ten years old [52]. Infancy BMI and BMI onward were also negatively associated with carotid distensibility (a measure of arterial elasticity) in healthy 10-year-old children, of which the underlying mechanism is unknown. However, this association may be mediated by the usual metabolic complications associated with obesity (inflammation, insulin resistance and high blood pressure) [52]. Similarly, BMI z-scores and abdominal adiposity measures such as waist-to-hip ratio were strong predictors of vascular function in children and adolescents [53]; however, childhood obesity as a risk factor on its own promotes low-grade inflammation by inducing the production of arginase, which will compete with eNOS, thereby inhibiting NO production [43].

Furthermore, maternal weight gain during pregnancy may also be a risk factor for increased risk of cardiovascular disease, regardless of the child’s age, as synthesized from a recent systematic review [54]. Therefore, maternal obesity before pregnancy and gestational weight gain increase the risk of childhood and adolescent hypertension, dyslipidemia, hyperglycemia and insulin resistance [54]. Combined with exposure to maternal obesity, exposure to maternal smoking and alcohol use may exacerbate these effects on cardiovascular risk in offspring.

In addition to adiposity indices, smoking during pregnancy may induce health consequences such as high blood pressure in offspring [1]. To illustrate, maternal smoking was not significantly associated with SBP in six-year-old children, but a dose–response association was reported with DBP [36]. In another study, compared to children with no exposure to tobacco smoking in utero, children exposed to continuous maternal smoking of more than five cigarettes per day presented higher SBP readings and increased risk of cardiometabolic risk factor clustering [31]. Finally, a recent systematic review and meta-analysis observed a significantly increased SBP (β = 0.31 mmHg) in studies on children exposed to maternal smoking during pregnancy [55]. However, solid conclusions could not be drawn due to the low level of evidence. Furthermore, a study on 6–9-year-old children in the Eastern Cape, South Africa, reported an association between obesity, elevated blood pressure, microalbuminuria and oxidative stress with vascular dysfunction in children [43]. Hypertension in premature individuals was also associated with endothelial dysfunction and changes in the renin–angiotensin–aldosterone (RAAS) system, the sympathetic nervous system and renal alterations [6]. In offspring exposed to alcohol exposure during pregnancy, only one study observed a significantly higher SBP, DBP and MAP in rat offspring [19]. Therefore, the detrimental effects of maternal smoking and alcohol exposure in utero on vascular function in offspring may be mediated through elevated systolic blood pressure. 

### 4.2. Additional Influences during Childhood

Several additional factors may further complicate the effect of tobacco smoking and alcohol exposure during pregnancy and influence the development of CVD risk factors in their offspring as they age [41]. These risk factors include environmental tobacco smoke exposure, poor dietary intake, pollution, physical inactivity and psychosocial stress [56,57,58,59,60]. For example, socioeconomically underprivileged individuals typically consume foods high in salt and saturated fats associated with hypertension [61]. A recent systematic review compared behavioral risk factors for non-communicable diseases (NCDs) in low and high socioeconomic groups among several countries [62]. Allen et al. (2017) reported that low socioeconomic groups in low-to-middle-income countries were more likely to smoke, consume alcohol and consume fewer fruits and vegetables, fiber and fish. In contrast, high socioeconomic groups had a higher prevalence of physical inactivity and consumed more processed foods, fats and salts [62]. 

In addition, the prospective, population-based Young Finns Study investigated risk factors in individuals from a high socioeconomic disadvantage measured by education, employment and home ownership from childhood (6–21 years) until adulthood (22–48 years). High socioeconomic disadvantage was characterized by reduced intake of fruits and vegetables from six years of age, decreased physical activity and increased tobacco smoking prevalence from the age of 12 years and increased fasting glucose from 27 years of age [63]. Interestingly, adults continuously exposed to a high socioeconomic disadvantage during their childhood had a worse cardiometabolic profile, such as obesity, a fatty liver, hypertension and a higher risk for diabetes [63]. Therefore, several factors may contribute to the increased risk of cardiometabolic risk factors in childhood, which can occur across multiple levels such as the individual itself, the family/parents, neighborhood/community and broader settings.

In addition to socioeconomic factors, there is evidence that environmental factors such as air pollution, noise pollution and second-hand smoke exposure may be associated with early markers of vascular dysfunction. For example, second-hand smoke exposure was shown to impair endothelial function, as shown with reduced FMD. Juonala et al. (2012) (as cited in US Surgeon General’s report, 2014) observed that brief exposure to second-hand smoke exposure of less than one hour results in reduced FMD and the release of von Willebrand factor antigen and endothelial progenitor cells in response to endothelial injury [64]. A review by Hahad et al. (2019) discussed the molecular effects of environmental noise on oxidative stress and vascular dysfunction. The evidence suggests that chronic exposure to noise on its own may influence hemodynamics, oxidative stress, vascular function, autonomic tone, and inflammation, increasing the risk of vascular dysfunction and cerebrocardiovascular disease. Hahad et al. and colleagues explain that long-term exposure can stimulate the sympathetic nervous system, causing a pathophysiological cascade of events, subsequently leading to increased heart rate, blood pressure and stress hormone levels [65]. This promotes the development of cerebrocardiovascular disease risk factors, leading to a stroke, myocardial infarction, heart failure or ischemic heart disease [65]. Therefore, environmental factors may play a mediating role in the development of vascular dysfunction and cardiovascular development.

In addition to noise pollution, air pollutants such as oxides of nitrogen, particulate matter and carbon monoxide have been linked to CVD risk factors, subclinical CVD and cardiovascular mortality [66]. In children, a study observed an association between long-term exposure to air pollution and increased blood pressure and hypertension, with stronger associations in obese children [59]. This means that overweight and obesity in children increase their susceptibility to the health effects of long-term exposure to ambient air pollution, such as sulfur dioxide, nitrogen dioxide and ozone concentrations.

### 4.3. Limitations

Research on vascular dysfunction in children exposed to maternal smoking and alcohol use in utero is scarce, with few studies focusing on vascular health, cholesterol and blood pressure. The sample size in animal studies was rather small compared to human studies. In addition, results in this review included studies performed on different animals, increasing possible heterogeneity among animal studies. In human studies, there is a limited number of articles in South Africa, with more studies conducted in the Netherlands and the United States, limiting the generalizability of the results to children in low socioeconomic populations. Therefore, the overall estimated effect of teratogen exposure on vascular structure and function is beyond the scope of this review, although a systematic review is currently in progress to confirm the certainty of all the evidence [67].

## 5. Conclusions

The effects of fetal exposure to alcohol and tobacco smoking on the fetus are complex, as their suggested pathophysiological pathways may occur simultaneously or subsequently. Despite its complexity, the literature suggests that reduction in NO bioavailability and alterations in the epigenome during in utero exposure through the upregulation of pro-oxidative and proinflammatory genes may be the common denominators. Evidence suggests that potential mediators, including birth size, altered lipid metabolism, childhood obesity and elevated blood pressure in offspring, may increase the risk of vascular dysfunction in childhood. However, many other environmental factors play a role which can occur across multiple levels, such as the individual itself, the family/parents, the neighborhood/community and broader settings. This challenges research and intervention, particularly on the potential compounding effects of multiple teratogens on the developing fetus. Because research is limited and the long-term effects of teratogen exposure, particularly alcohol, on offspring vascular structure and function remain uncertain, studies on both maternal alcohol and cigarette use during pregnancy are needed, specifically in a South African context. 

## Figures and Tables

**Figure 1 ijerph-20-06398-f001:**
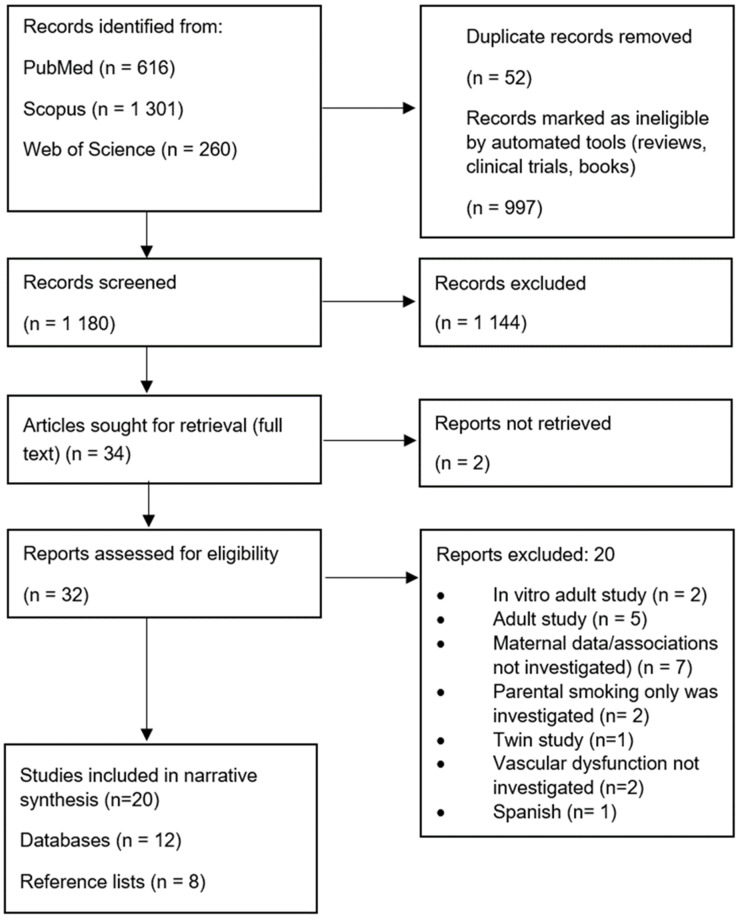
Flow chart of study selection for narrative synthesis.

**Figure 2 ijerph-20-06398-f002:**
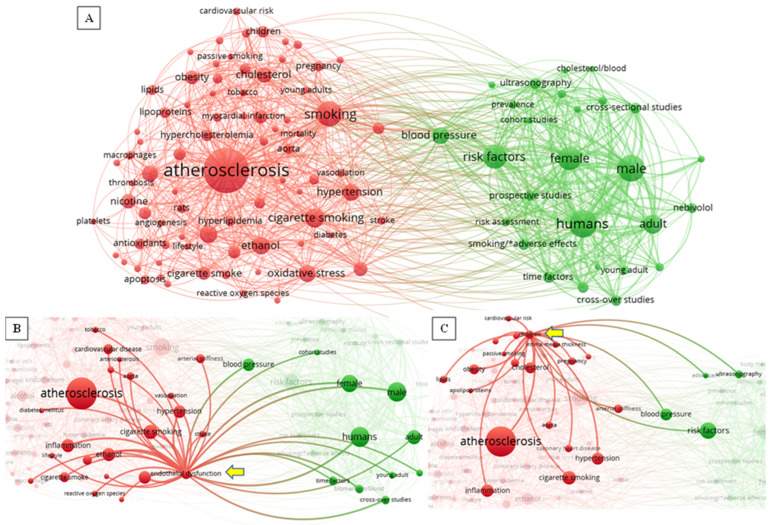
(**A**). A graphical network generated in VOSviewer™ shows the commonality of keywords in the literature search for this review. (**B**). A network showing the most common terms co-occurring with the term “endothelial dysfunction” indicated by the yellow arrow. (**C**). A network showing the most common terms co-occurring with the keyword “children” indicated by the yellow arrow.

**Figure 3 ijerph-20-06398-f003:**
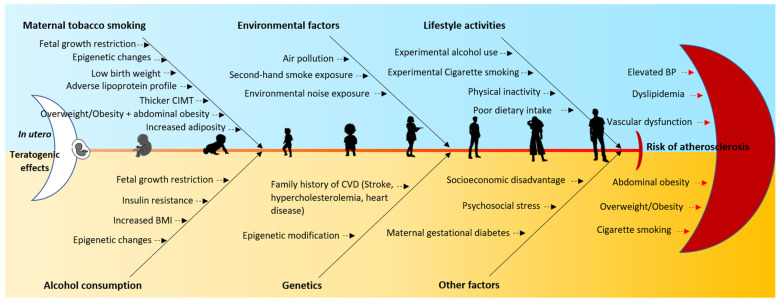
Fishbone diagram summarizing the effects of maternal cigarette smoking and alcohol consumption on cardiometabolic risk in children and several factors throughout childhood that may exacerbate these effects in the development of cardiometabolic risk factors and vascular dysfunction.

## Data Availability

All data generated and summarized during this narrative review are included in the published review article.

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
