# Peer review of "Vascular Effects, Potential Pathways and Mediators of Fetal Exposure to Alcohol and Cigarette Smoking during Pregnancy: A Narrative Review"

_ijerph, 2023, doi:10.3390/ijerph20146398_

Round 1
Reviewer 1 Report
Nice work and good efforts explaining the potential hazards of teratogens as alcohol and tobacco pl , however references need to be checked and corrected due to minor mistakes as that in reference number 13,21,37
Reviewer 2 Report
Tammy C Hartel et al. write the paper “Vascular effects, potential pathways and mediators of fetal exposure to alcohol and cigarette smoking during pregnancy: A narrative review”.
This is a review on fetal alcohol and cigarette smoke exposure and vascular and metabolic effects.
The method used to perform the review is correct and is based on various bibliographic sources and uses softwares that select papers based on keywords.
The literature presents evidence supporting the detrimental effects of fetal exposure to both alcohol and tobacco smoking on vascular alterations in both human and animal studies.
These xenobiotics may be the cause of atherosclerosis by: reduction in nitric oxide (NO) bioavailability, alterations in the epigenome in infants and in proinflammatory genes.
The authors conclude that maternal smoking and alcohol consumption have more negative outcomes on the infant in the short term, while several factors during the first few years of life may mediate the development of vascular dysfunction.
The abstract describes the study well. However when the authors write: “The literature presents evidence supporting the detrimental effects of fetal exposure to both alcohol and tobacco smoking on vascular alterations in both human and animal studies” it does not seem to me a valid description, because human studies are all after exposure to tobacco smoking and only 1 study after exposure to both tobacco smoking and ethanol. Furthermore seven animal studies reguard exposure to ethanol and three to nicotine, which is not the same as exposure to tobacco smoking.
The background describes the effects of tobacco smoking on the cardiovascular system but does not describe the effects of ethanol. This part should be supplemented.
The materials and methods are comprehensive and written in an understandable way. Human studies are all prospective and this is a positive fact. Animal studies have been performed on different animals.
The results clearly report the negative effects of tobacco smoking and ethanol on the human cardiovascular system and lipid metabolism both in animal and human studies.
In the discussion, the authors synthesize the most recent evidence on the effects of fetal exposure to maternal tobacco smoking and alcohol use and vascular dysfunction in offspring. However, while prospective studies in humans after tobacco smoking exposure are avaliable, data about ethanol exposure effects are lacking.
The authors conclude that studies of maternal alcohol and cigarette use during pregnancy are needed, particularly in a South African context.
Minor Revisions:
11. Line 11, Abstract: Background: Programming of atherosclerosis results in vascular structure… should write: Abstract: 1) Background: Programming of atherosclerosis results in vascular structure;
2. Tables 1 and 2 contain acronyms not explained in the tables:
3. The caption of Fig. 1 is missing;
4. The software used should be mentioned in the caption of Fig. 2;
5. Fig. 3 has the caption inside the figure;
6. References 4, 16, 25, 30, 37 (to be controlled), 41, 56 (to be controlled), 61 DOI must be addet
The paper may be accepted after minor revision.
Reviewer 3 Report
Dear Authors
Your submission described one of the most critical problems in maternal- fetal medicine. It is also a social problem.
But the essential sequels on childhood, and adulthood of offspring in pregnancies complicated by maternal abuse of nicotine or alcohol are related to placental velocities, inflammation, and immunology. In Your research placental pathology and its relation to the epigenetic model of these pregnancies is missed. The abuse of nicotine as well as alcohol is related to obstetrics complications, prematurity, and delayed neurological development of offspring.
Dear Authors
I think that Your submission must require minor editing of the English language.
Round 2
Reviewer 3 Report
Dear Authors
In this form Your submission is acceptable for publishing in an eminent Journal as IJERPH.
Sincerely
Ivana Babovic MD Ph.D Associated Professor
Faculty of Medicine, University of Belgrade, Serbia